# Human Coronaviruses and Other Respiratory Viruses: Underestimated Opportunistic Pathogens of the Central Nervous System?

**DOI:** 10.3390/v12010014

**Published:** 2019-12-20

**Authors:** Marc Desforges, Alain Le Coupanec, Philippe Dubeau, Andréanne Bourgouin, Louise Lajoie, Mathieu Dubé, Pierre J. Talbot

**Affiliations:** 1Laboratory of Neuroimmunovirology, Institut national de la recherche scientifique (INRS)-Institut Armand-Frappier, Université du Québec, Laval, QC H7V 1B7, Canada; alainl2258@hotmail.fr (A.L.C.); ppedubeau@gmail.com (P.D.); Andreanne.Bourgouin@uqtr.ca (A.B.);; 2Faculté de médecine et des sciences de la santé, Université de Sherbrooke, Sherbrooke, QC J1K 2R1, Canada; louise.lajoie@gmail.com; 3Research Centre of the Centre Hospitalier de l’Université de Montréal (CRCHUM), Montreal, QC H3T 1J4, Canada

**Keywords:** human respiratory virus, human coronavirus, respiratory viral infection, neuroinvasion, CNS infection, acute and chronic neurological diseases, encephalitis, encephalopathy

## Abstract

Respiratory viruses infect the human upper respiratory tract, mostly causing mild diseases. However, in vulnerable populations, such as newborns, infants, the elderly and immune-compromised individuals, these opportunistic pathogens can also affect the lower respiratory tract, causing a more severe disease (e.g., pneumonia). Respiratory viruses can also exacerbate asthma and lead to various types of respiratory distress syndromes. Furthermore, as they can adapt fast and cross the species barrier, some of these pathogens, like influenza A and SARS-CoV, have occasionally caused epidemics or pandemics, and were associated with more serious clinical diseases and even mortality. For a few decades now, data reported in the scientific literature has also demonstrated that several respiratory viruses have neuroinvasive capacities, since they can spread from the respiratory tract to the central nervous system (CNS). Viruses infecting human CNS cells could then cause different types of encephalopathy, including encephalitis, and long-term neurological diseases. Like other well-recognized neuroinvasive human viruses, respiratory viruses may damage the CNS as a result of misdirected host immune responses that could be associated with autoimmunity in susceptible individuals (virus-induced neuro-immunopathology) and/or viral replication, which directly causes damage to CNS cells (virus-induced neuropathology). The etiological agent of several neurological disorders remains unidentified. Opportunistic human respiratory pathogens could be associated with the triggering or the exacerbation of these disorders whose etiology remains poorly understood. Herein, we present a global portrait of some of the most prevalent or emerging human respiratory viruses that have been associated with possible pathogenic processes in CNS infection, with a special emphasis on human coronaviruses.

## 1. Introduction

The central nervous system (CNS), a marvel of intricate cellular and molecular interactions, maintains life and orchestrates homeostasis. Unfortunately, the CNS is not immune to alterations that lead to neurological disease, some resulting from acute, persistent or latent viral infections. Several viruses have the ability to invade the CNS, where they can infect resident cells, including the neurons [1]. Although rare, viral infections of the CNS do occur [2]. However, their incidence in clinical practice is difficult to evaluate precisely [3]. For instance, in cases of viral encephalitis involving the most prevalent viruses known to reach the CNS (mainly herpesviruses, arboviruses and enteroviruses), an actual viral presence can only be detected in 3 to 30 cases out of 100,000 persons. Considering all types of viral infections, between 6000 and 20,000 cases of encephalitis that require hospitalization occur every year in the United States, representing about 6 cases per 100,000 infected persons every year. As the estimated charge for each case lies between $58,000 and $89,600, an evaluation of the total annual health cost is of half a billion dollars [1,4]. Due to the cost associated with patient care and treatment, CNS viral infections cause considerably more morbidity and disabilities in low-income/resource-poor countries [5,6].

Very common worldwide, viral infections of the respiratory tract represent a major problem for human and animal health, imposing a tremendous economic burden. These respiratory infections induce the most common illnesses [7] and are a leading cause of morbidity and mortality in humans worldwide, causing critical problems in public health, especially in children, the elderly and immune-compromised individuals [8,9,10,11,12]. Viruses represent the most prevalent pathogens present in the respiratory tract. Indeed, it is estimated that about 200 different viruses (including influenza viruses, coronaviruses, rhinoviruses, adenoviruses, metapneumoviruses, such as human metapneumovirus A1, as well as orthopneumoviruses, such as the human respiratory syncytial virus) can infect the human airway. Infants and children [13], as well as the elderly represent more vulnerable populations, in which viruses cause 95% and 40% of all respiratory diseases, respectively [11]. Among the various respiratory viruses, some are constantly circulating every year in the human populations worldwide, where they can be associated with a plethora of symptoms, from common colds to more severe problems requiring hospitalization [8,14,15,16,17,18,19,20]. Moreover, in addition to the many “regular” viruses that circulate and infect millions of people every year, new respiratory viral agents emerge from time to time, causing viral epidemics or pandemics associated with more serious symptoms [21,22], such as neurologic disorders. These peculiar events usually take place when RNA viruses like influenza A, human coronaviruses, such as MERS-CoV and SARS-CoV, or henipaviruses, present in an animal reservoir, cross the species barrier as an opportunistic strategy to adapt to new environments and/or new hosts. These zoonoses may have disastrous consequences in humans [23,24,25,26,27,28,29], and the burden is even higher if they have neurological consequences.

## 2. Viruses as Plausible Etiologic Agents in Neurological Disorders

Epithelial cells that line the respiratory tract are the first cells that can be infected by respiratory viruses. Most of these infections are self-limited and the virus is cleared by immunity with minimal clinical consequences. On the other hand, in more vulnerable individuals, viruses can also reach the lower respiratory tract where they cause more severe illnesses, such as bronchitis, pneumonia, exacerbations of asthma, chronic obstructive pulmonary disease (COPD) and different types of severe respiratory distress syndromes [15,16,18,19,20,30,31]. Besides all these respiratory issues, accumulating evidence from the clinical/medical world strongly suggest that, being opportunistic pathogens, these viruses are able to escape the immune response and cause more severe respiratory diseases or even spread to extra-respiratory organs, including the central nervous system [8,32,33] where they could infect resident cells and potentially induce other types of pathologies [7,34,35].

Like all types of viral agents, respiratory viruses may enter the CNS through the hematogenous or neuronal retrograde route. In the first, the CNS is being invaded by a viral agent which utilizes the bloodstream and in the latter, a given virus infects neurons in the periphery and uses the axonal transport machinery to gain access to the CNS [35,36,37,38,39]. In the hematogenous route, a virus will either infect endothelial cells of the blood-brain-barrier (BBB) or epithelial cells of the blood-cerebrospinal fluid barrier (BCSFB) in the choroid plexus (CP) located in the ventricles of the brain, or leukocytes that will serve as a vector for dissemination towards the CNS. Viruses such as HIV [40,41,42,43], HSV [44,45], HCMV [46,47], enteroviruses such as coxsackievirus B3 [48,49], flaviviruses [50], chikungunya virus (CHIKV) [51] and echovirus 30 [52] have all been shown to disseminate towards the CNS through the hematogenous route. Respiratory viruses such as RSV [8,53,54], henipaviruses [55,56], influenza A and B [57,58,59] and enterovirus D68 [60] are also sometimes found in the blood and, being neuroinvasive, they may therefore use the hematogenous route to reach the CNS. As they invade the human host through the airway, the same respiratory viruses may use the olfactory nerve to get access to the brain through the olfactory bulb [35,36,61,62,63]. On the other hand, these viruses may also use other peripheral nerves like the trigeminal nerve, which possesses nociceptive neuronal cells present in the nasal cavity [64,65], or alternatively, the sensory fibers of the vagus nerve, which stems from the brainstem and innervates different organs of the respiratory tract, including the larynx, the trachea and the lungs [8,66,67,68,69].

Although the CNS seems difficult for viruses to penetrate, those pathogens that are able to do so may disseminate and replicate very actively and will possibly induce an overreacting innate immune response, which may be devastating. This situation may lead to severe meningitis and encephalitis that can be fatal, depending on several viral and host factors (including immunosuppression due to disease or medications) that may influence the severity of the disease. 

Recently, a very interesting manuscript produced by Bookstaver and collaborators underlined the difficulties of precisely deciphering the epidemiology and identifying the causal agent of CNS infections [70]. These difficulties are mainly due to the tremendous variation in the symptoms throughout the disease process and to the myriad of viruses that can cause CNS infections. As stated in their report, these authors underlined that the clinical portrait of viral infections is often nonspecific and requires the clinician to consider a range of differential diagnoses. Meningitis (infection/inflammation in meninges and the spinal cord) produces characteristic symptoms: fever, neck stiffness, photophobia and/or phonophobia. Encephalitis (infection/inflammation in the brain and surrounding tissues) may remain undiagnosed since the symptoms may be mild or non-existent. Symptoms may include altered brain function (altered mental status, personality change, abnormal behavior or speech), movement disorders and focal neurologic signs, such as hemiparesis, flaccid paralysis or paresthesia. Seizures can occur during both viral meningitis and encephalitis [70]. Furthermore, viral encephalitis may also be difficult to distinguish from a non-viral encephalopathy or from an encephalopathy associated with a systemic viral infection occurring outside the CNS. Considering all these observations, it is therefore mandatory to insist on the importance of investigating the patient’s history before trying to identify a specific viral cause of a given neurological disorder [70,71,72,73].

In humans, a long list of viruses may invade the CNS, where they can infect the different resident cells (neuronal as well as glial cells) and possibly induce or contribute to neurological diseases [74], such as acute encephalitis, which can be from benign to fatal, depending on virus tropism [75], pathogenicity as well as other viral and patient characteristics. For instance, 30 years ago, the incidence of children encephalitis was as high as 16/100,000 in the second year of life, while progressively reducing to 1/100,000 by the age of 15 [76]. More recent data indicate that, in the USA, the herpes simplex virus (HSV) accounts for 50–75% of identified viral encephalitis cases, whereas the varicella zoster virus (VZV), enteroviruses and arboviruses are responsible for the majority of the other cases in the general population [4]. Several other viruses can induce short-term neurological problems. For example, the rabies virus [77], herpes simplex and other herpes viruses (HHV) [72,78,79,80,81], arthropod-borne flaviviruses such as the West Nile virus (WNV), Japanese encephalitis virus (JEV), chikungunya virus (CHIKV), Zika virus (ZIKV), alphaviruses such as the Venezuelan, Western and Eastern equine encephalitis viruses [4,50,82,83,84,85,86,87] and enteroviruses [48,88] affect millions of individuals worldwide and are sometimes associated with encephalitis, meningitis and other neurological disorders. The presence of viruses in the CNS may also result in long-term neurological diseases and/or sequelae. Human immunodeficiency virus (HIV) induces neurodegeneration [89,90,91], which lead to motor dysfunctions and cognitive impairments [92]. Progressive multifocal leukoencephalopathy (PML) is a demyelinating disease [93] associated with reactivation of latent polyoma JC virus (JCV) [94,95]. Progressive tropical spastic paraparesis/HTLV-1-associated myelopathy (PTSP/HAM) is caused by human T-lymphotropic virus (HTLV-1) in 1–2% of infected individuals [95,96]. Measles virus (MV), a highly contagious common virus, is associated with febrile illness, fever, cough and congestion [97,98], as well as a characteristic rash and Koplik’s spots [99]. In rare circumstances, significant long-term CNS diseases, such as [99] post-infectious encephalomyelitis (PIE) or acute disseminated encephalomyelitis (ADEM), occur in children and adolescents. Other examples of rare but devastating neurological disorders are measles inclusion body encephalitis (MIBE), mostly observed in immune-compromised patients, and subacute sclerosing panencephalitis (SSPE) that appears 6–10 years after infection [100].

Yet, with the exception of HIV, no specific virus has been constantly associated with specific human neurodegenerative disease. On the other hand, different human herpes viruses have been associated with Alzheimer’s disease (AD), multiple sclerosis (MS) and other types of long-term CNS disorders [101,102,103]. As accurately stated by Majde [104], long-term neurodegenerative disorders may represent a “hit-and-run” type of pathology, since some symptoms are triggered by innate immunity associated with glial cell activation. Different forms of long-term sequelae (cognitive deficits and behavior changes, decreased memory/learning, hearing loss, neuromuscular outcomes/muscular weakness) were also observed following arboviral infections [83,103,105,106,107].

Including the few examples listed above, more than one hundred infectious agents (much of them being viruses) have been described as potentially encephalitogenic and an increasing number of positive viral identifications are now made with the help of modern molecular diagnostic methods [8,70,108,109,110]. However, even after almost two decades into the 21st century and despite tremendous advances in clinical microbiology, the precise cause of CNS viral infections often remains unknown. Indeed, even though very important technical improvements were made in the capacity to detect the etiological agent, identification is still not possible in at least half of the cases [110,111]. Among all the reported cases of encephalitis and other encephalopathies and even neurodegenerative processes, respiratory viruses could represent an underestimated part of etiological agents [104].

## 3. Respiratory Viruses with Neuroinvasive and Neurotropic Properties: Possible Associated Neuropathologies

Respiratory syncytial virus (RSV), a member of the *Orthopneumovirus* genus [112], infects approximately 70% of infants before the age of 1 and almost 100% by the age of 2 years old [113], making it the most common pathogen to cause lower respiratory tract infection such as bronchiolitis and pneumonia in infants worldwide [32,114]. Recent evidence also indicates that severe respiratory diseases related to RSV are also frequent in immunocompromised adult patients [8,115] and that the virus can also present neuroinvasive properties [8]. Over the last five decades, a number of clinical cases have potentially associated the virus with CNS pathologies. RSV has been detected in the cerebrospinal fluid (CSF) of patients (mainly infants) and was associated with convulsions, febrile seizures and different types of encephalopathy, including clinical signs of ataxia and hormonal problems [116,117,118,119,120,121,122,123,124,125,126]. Furthermore, RSV is now known to be able to infect sensory neurons in the lungs and to spread from the airways to the CNS in mice after intranasal inoculation, and to induce long-term sequelae such as behavioral and cognitive impairments [127].

An additional highly prevalent human respiratory pathogen with neuroinvasive and neurovirulent potential is the human metapneumovirus (hMPV). Discovered at the beginning of the 21st century in the Netherlands [128], it mainly causes respiratory diseases in newborns, infants and immunocompromised individuals [129]. During the last two decades, sporadic cases of febrile seizures, encephalitis and encephalopathies (associated with epileptic symptoms) have been described. Viral material was detected within the CNS in some clinical cases of encephalitis/encephalopathy [130,131,132,133,134] but, at present, no experimental data from any animal model exist that would help to understand the underlying mechanism associated with hMPV neuroinvasion and potential neurovirulence.

Hendra virus (HeV) and Nipah virus (NiV) are both highly pathogenic zoonotic members of the *Henipavirus* genus and represent important emerging viruses discovered in the late 1990s in Australia and southern Asia. They are the etiological agents of acute and severe respiratory disease in humans, including pneumonia, pulmonary edema and necrotizing alveolitis with hemorrhage [135,136,137,138]. Although very similar at the genomic level, both viruses infect different intermediate animal reservoirs: the horse for HeV and the pig for NiV as a first step before crossing the barrier species towards humans [135]. In humans, it can lead to different types of encephalitis, as several types of CNS resident cells (including neurons) can be infected [139,140]. The neurological signs can include confusion, motor deficits, seizures, febrile encephalitic syndrome and a reduced level of consciousness. Even neuropsychiatric sequelae have been reported but it remains unclear whether a post-infectious encephalo-myelitis occurs following infection [141,142,143]. The use of animal models showed that the main route of entry into the CNS is the olfactory nerve [144] and that the Nipah virus may persist in different regions of the brain of grivets/green monkeys [145], reminiscent of relapsing and late-onset encephalitis observed in humans [146].

Influenza viruses are classified in four types: A, B, C and D. All are endemic viruses with types A and B being the most prevalent and causing the flu syndrome, characterized by chills, fever, headache, sore throat and muscle pain. They are responsible for seasonal epidemics that affect 3 to 5 million humans, among which 500,000 to 1 million cases are lethal each year [147,148]. Associated with all major pandemics since the beginning of the 20th century, circulating influenza A presents the greatest threat to human health. Most influenza virus infections remain confined to the upper respiratory tract, although some can lead to severe cases and may result in pneumonia, acute respiratory distress syndrome (ARDS) [30,149] and complications involving the CNS [150,151,152]. Several studies have shown that influenza A can be associated with encephalitis, Reye’s syndrome, febrile seizure, Guillain–Barré syndrome, acute necrotizing encephalopathy and possibly acute disseminated encephalomyelitis (ADEM) [153,154,155,156,157,158]. Animal models have shown that, using either the olfactory route or vagus nerve, influenza A virus may have access to the CNS and alter the hippocampus and the regulation of neurotransmission, while affecting cognition and behavior as long-term sequelae [8,69,159,160,161,162]. The influenza A virus has also been associated with the risk of developing Parkinson’s disease (PD) [151] and has recently been shown to exacerbate experimental autoimmune encephalomyelitis (EAE), which is reminiscent of the observation that multiple sclerosis (MS) relapses have been associated with viral infections (including influenza A) of the upper respiratory tract [163,164,165]. 

Another source of concern when considering human respiratory pathogens associated with potential neuroinvasion and neurovirulence is the *Enterovirus* genus, which comprises hundreds of different serotypes, including polioviruses (PV), coxsackieviruses (CV), echoviruses, human rhinoviruses (HRV) and enteroviruses (EV). This genus constitutes one of the most common cause of respiratory infections (going from common cold to more severe illnesses) and some members (PV, EV-A71 and -D68, and to a lesser extent HRV) can invade and infect the CNS, with detrimental consequences [166,167,168,169]. Even though extremely rare, HRV-induced meningitis and cerebellitis have been described [170]. Although EV infections are mostly asymptomatic, outbreaks of EV-A71 and D68 have also been reported in different parts of the world during the last decade. EV-A71 is an etiological agent of the hand–foot–mouth disease (HFMD) and has occasionally been associated with upper respiratory tract infections. EV-D68 causes different types of upper and lower respiratory tract infections, including severe respiratory syndromes [171]. Both serotypes have been associated with neurological disorders like acute flaccid paralysis (AFP), myelitis (AFM), meningitis and encephalitis [166,172,173,174,175].

Last but not least, human coronaviruses (HCoV) are another group of respiratory viruses that can naturally reach the CNS in humans and could potentially be associated with neurological symptoms. These ubiquitous human pathogens are molecularly related in structure and mode of replication with neuroinvasive animal coronaviruses [176] like PHEV (porcine hemagglutinating encephalitis virus) [177], FCoV (feline coronavirus) [178,179] and the MHV (mouse hepatitis virus) strains of MuCoV [180], which can all reach the CNS and induce different types of neuropathologies. MHV represents the best described coronavirus involved in short- and long-term neurological disorders (a model for demyelinating MS-like diseases) [181,182,183]. Taken together, all these data bring us to consider a plausible involvement of HCoV in neurological diseases.

## 4. Human Coronaviruses: Eclosions of Recognized Respiratory Pathogens

The first strains of HCoV were isolated in the mid-60s from patients presenting an upper respiratory tract disease [184,185,186,187]. Before the Severe Acute Respiratory Syndrome (SARS) appeared in 2002 and was associated with SARS-CoV [188,189,190], only two groups of HCoV, namely HCoV-229E (previous group 1, now classified as *Alphacoronavirus*) and HCoV-OC43 (previous group 2, now classified as *Betacoronavirus*) were known. Several new coronaviruses have now been identified, including three that infect humans: alphacoronavirus HCoV-NL63 [191] and betacoronaviruses HCoV-HKU1 and MERS-CoV [192,193]. 

The HCoV-229E, -OC43, -NL63 and -HKU1 strains are endemic worldwide [31,184,194,195,196,197,198,199] and exist in different genotypes [200,201,202,203,204,205,206,207]. In immunocompetent individuals they usually infect the upper respiratory tract, where they are mainly associated with 15–30% of upper respiratory tract infections (URI): rhinitis, laryngitis/pharyngitis as well as otitis. Being highly opportunistic pathogens [14], HCoV can reach the lower respiratory tract and be associated with more severe illnesses, such as bronchitis, bronchiolitis, pneumonia, exacerbations of asthma and respiratory distress syndrome [17,31,208,209,210,211,212,213,214].

The 2002–2003 SARS pandemic was caused by a coronavirus that emerged from bats (first reservoir) [25] to infect palm civets (intermediary reservoir) and then humans [215]. A total of 8096 probable cases were reported and almost 10% (774 cases in more than 30 countries) of these resulted in death [216,217,218]. The clinical portrait was described as an initial flu-like syndrome, followed by a respiratory syndrome associated with cough and dyspnea, complicated with the “real” severe acute respiratory syndrome (SARS) in about 20% of the patients [31,219]. In addition, multiple organ failure was observed in several SARS-CoV-infected patients [220]. 

In the fall of 2012, individuals travelling from the Arabian Peninsula to the United Kingdom were affected by the Middle-East Respiratory Syndrome (MERS), a severe lower respiratory tract infection that resembled SARS, leading also to gastrointestinal symptoms and renal failure among some patients [221]. Molecular sequencing rapidly showed that the new epidemic was caused by a new coronavirus: the MERS-CoV [193,222,223]. MERS-CoV most probably originated from bats before infecting an intermediary reservoir (the dromedary camel), and also represented a zoonotic transmission to humans. Phylogenetic analyses suggest that there have been multiple independent zoonotic introductions of the virus in the human population. Moreover, nosocomial transmission was observed in multiple hospitals in Saudi Arabia [221,224,225,226,227,228,229,230]. 

Although possible, human-to-human MERS-CoV transmission appears inefficient as it requires extended close contact with an infected individual. Consequently, most transmission have occurred among patients’ families and healthcare workers (clusters of transmission). A more efficient human-to-human transmission was observed in South Korea, during the 2015 outbreak of MERS-CoV [231,232]. Even though it has propagated to a few thousand people and possesses a high degree of virulence, MERS-CoV seems mostly restricted to the Arabic peninsula and is not currently considered an important pandemic threat. However, virus surveillance and better characterization are warranted, in order to be prompt to respond to any change in that matter [23,233,234,235].

As of October 8, 2019, the World Health Organization (WHO) reported that MERS-CoV had spread to at least 27 different countries, where 2468 laboratory-confirmed human cases have been identified with 851 being fatal (https://www.who.int/emergencies/mers-cov/en/). As observed for the four circulating strains of HCoV [31,194], both SARS-CoV and MERS-CoV usually induce more [228,236] severe illnesses, and strike stronger in vulnerable populations such as the elderly, infants, immune-compromised individuals or patients with comorbidities [31,237].

### 4.1. Human Coronaviruses: Pathogens Causing Illnesses Outside the Respiratory Tract

Over the years, like SARS- and MERS-CoV, the four endemic HCoV have also been identified as possible etiological agents for pathologies outside the respiratory tract. Indeed, myocarditis, meningitis, severe diarrhea (and other gastrointestinal problems) and multi-organ failure [220,221,238,239,240,241] have been reported, especially in children. Recent investigations on HCoV as enteric pathogens demonstrated that all HCoV strains can be found in stool samples of children with acute gastroenteritis; however, no evidence of association could yet be clearly demonstrated with disease etiology [242,243]. Different reports also presented a possible link between the presence of HCoV within the human central nervous system (CNS) and some neurological disorders among patients examined [244,245,246,247,248,249]. Like all viruses, HCoV may enter the CNS through the hematogenous or neuronal retrograde route.

#### 4.1.1. Possible Mechanisms of HCoV Neuroinvasiveness

In the human airways, HCoV infection may lead to the disruption of the nasal epithelium [250] and, although they bud and are released mostly on the apical side of the epithelial cells, a significant amount of viruses is also released from the basolateral side [251]. Thus, although HCoV infections are, most of the time, restricted to the airways, they may under poorly understood conditions pass through the epithelium barrier and reach the bloodstream or lymph and propagate towards other tissues, including the CNS [33,38,208,252]; this was also suggested for other respiratory viruses that can reach the human CNS, namely, RSV [8,53], Nipah virus [55] and influenza virus [57,58,59].

Different strains of HCoV, including SARS-CoV, can infect different myeloid cells [14,220,252,253,254,255,256] to manipulate the innate immunity and to disseminate to other tissues, including the CNS, where they may be associated with other type of pathologies, especially in immunocompromised individuals as it was observed for murine cytomegalovirus (MCMV) [257,258]. Moreover, persistently-infected leukocytes [252] may serve as a reservoir and vector for neuroinvasive HCoV [245]. Therefore, neuroinvasive HCoV could use the hematogenous route to penetrate into the CNS. 

The second form of any viral spread towards the CNS is through neuronal dissemination, where a given virus infects neurons in the periphery and uses the machinery of active transport within those cells in order to gain access to the CNS [35,36]. Although the olfactory bulb is highly efficient at controlling neuroinvasion, several viruses have been shown to enter CNS through the olfactory route [259,260]. After an intranasal infection, both HCoV-OC43 and SARS-CoV were shown to infect the respiratory tract in mice and to be neuroinvasive [261,262,263,264,265]. Over the years, we and others have gathered data showing that HCoV-OC43 is naturally neuroinvasive in both mice and humans [244,245,261,263,266]. Experimental intranasal infections of susceptible mice also indicate that, once it has invaded the CNS, the virus disseminated to several regions of the brain and the brainstem before it eventually reaches the spinal cord [266,267,268]. Furthermore, based on more recent work [269], Figure 1 illustrates the olfactory route, which is clearly the main route of neuroinvasion used by HCoV-OC43, as well as the early steps of subsequent neuropropagation within the CNS in susceptible mice and recapitulates the suggested equivalent pathway in humans. Nevertheless, our data suggest that HCoV-OC43 may also invade the CNS from the external environment through other pathways involving other cranial peripheral nerves [269], reminiscent of what was shown for other human respiratory viruses such as RSV and influenza virus [8]. Therefore, on the one hand, an apparently innocuous human respiratory pathogen such as the HCoV may reach the CNS by different routes and induce short-term illnesses, such as encephalitis. On the other hand, it may persist in resident cells of the human CNS and may become a factor or co-factor of neuropathogenesis associated with long-term neurological sequelae in genetically or otherwise predisposed individuals.

#### 4.1.2. Human Coronaviruses in the CNS: Possible Associated Neurological Pathologies

Because of their natural neuroinvasive potential in humans and animals, a possible association between the presence of ubiquitous human coronaviruses in the triggering or exacerbation of neurological human pathologies has often been suggested over the years. It is now accepted that HCoV are not always confined to the upper respiratory tract and that they can invade the CNS [220,245,248,270]. As other viruses listed herein, HCoV are neurotropic and potentially neurovirulent. Even though no clear cause and effect link has ever been made with the onset of human neurological diseases, their neuropathogenicity is being increasingly recognized in humans, as several recent reports associated cases of encephalitis [244], acute flaccid paralysis [271] and other neurological symptoms, including possible complications of HCoV infection such as Guillain–Barré syndrome or ADEM [249,272,273,274,275,276,277,278,279]. The presence and persistence of HCoV in human brains was proposed to cause long-term sequelae related to the development or aggravation of chronic neurological diseases [245,246,247,248,280,281,282]. Given their high prevalence [31,283], long-term persistence and newly recognized neuropathogenesis, HCoV disease burden could currently be underestimated. This suggest that better surveillance, diagnoses and deepened virus–host interactions studies are warranted in order to gather more knowledge that will make possible the development of therapeutic strategies to prevent or treat occurrences.

##### Potential Short-Term Neuropathologies

SARS-CoV, HCoV-OC43 and -229E are naturally neuroinvasive and neurotropic in humans and therefore potentially neurovirulent [220,244,245,249,270,271]. Furthermore, animal models showed that SARS-CoV could invade the CNS primarily through the olfactory route [265] or even after an intra-peritoneal infection [284], and induce neuronal cell death [265,284]. To our knowledge, no reports on the presence of the three other coronaviruses that infect humans in the CNS have been published. However, neurological symptoms have been described in patients infected by all three viruses [276,277,285].

Making use of our in vivo model of HCoV neuropathogenesis, relying on the natural susceptibility of mice to HCoV-OC43—the most prevalent strain among endemic HCoV [210,286]—encephalitis and transient flaccid paralysis associated with propagation towards the spinal cord and demyelination and long-term persistence in surviving mice were observed [261,266,267,268,287,288,289]; thus, recapitulating the neurological afflictions reported in some patients infected by HCoV [244,249,271,272,275,276,277,290]. Although we must interpret data obtained in rodents with all the caution dictated by the use of a non-human host, it is likely that the underlying mechanisms described will have relevance to the human situation or at least provide leads to investigate neurotropic HCoV in humans.

In susceptible mice, HCoV-OC43 has a selective tropism for neurons in which it is able to use axonal transport as a way of neuron-to-neuron propagation [269]. These results, together with data harvested with the use of microfluidic devices (Xona microfluidic), helped to elaborate a putative model of propagation adapted from Tomishima and Enquist [291], in which infectious HCoV-OC43 could either be assembled in the cell body or at different points along the axon using the anterograde axonal transport to propagate between neurons or from neurons to glial cells surrounding neurons in the CNS (Figure 2). Furthermore, based on previous data using different mutant recombinant viruses harboring mutation in the S protein [266,268,269] and making use of a luciferase expressing recombinant HCoV-OC43 [292,293,294], we are now showing that the rate and success of virus propagation towards the spinal cord, in part through the neuron-to-neuron pathway, correlates with the exacerbation of neurovirulence (Figure 3).

HCoV-OC43 structural and accessory proteins are important for infection and some clearly represent virulence factors [38,266,267,268,269,289,295,296]. Using neuronal cell cultures and our murine model, we gathered data indicating that some of these proteins also play a significant role in viral dissemination [269,296] and now aim to exploit these promising leads to fully understand the course and determinants of propagation to and through the CNS and complete the neurologic portrait of short term HCoV neuropathogenesis. 

##### Potential Long-Term Neuropathologies and Sequelae

The presence of HCoV RNA in the human CNS establishes the natural neuroinvasive properties of these respiratory viral agents. Moreover, it also suggests that they persist in human CNS [245] as they do in human neural cells [297,298] and in the CNS of mice that survive acute encephalitis. These surviving mice exhibited long-term sequelae associated with decreased activity in an open field test and a reduced hippocampus with neuronal loss in the CA1 and CA3 layers [287], reminiscent of what was observed after infection by the influenza A virus and RSV [127,162] and to the significant loss of synapses within the CA3 region after infection by WNV [299,300]. 

The precise and complete etiology of several long-term neurological pathologies still represent a conundrum. Multiple sclerosis (MS) represents one such neurological disease for which an infectious agent or agents may play a triggering role, with viruses the most likely culprit in genetically predisposed individuals [301]. It has been suggested that several neurotropic viruses could be involved in MS pathogenesis but that they may do so through similar direct and/or indirect mechanisms [302,303,304,305,306]. However, although research has not yet led to a direct link to any specific virus, association of coronaviruses with MS has been suggested [307]. Even though HCoV-OC43 and -229E were detected in some control brains and in some brains coming from patients with different neurological diseases, there was a significantly higher prevalence of HCoV-OC43 in brains of MS patients [245]. Moreover, autoreactive T cells were able to recognize both viral and myelin antigens in MS patients but not in controls during infection by HCoV-OC43 and HCoV-229E [308,309]. Thus, the immune response may participate in the induction or exacerbation of long-term neuropathologies such as MS in genetically or otherwise susceptible individuals. Furthermore, it was shown that in recombination activation gene (RAG) knock-out mice, HCoV-OC43-induced encephalitis could be partially mediated by the T-cell response to infection [263]. This underlines the possibility that, like its murine counterpart MHV, long term infection of the CNS by HCoV [245] may participate in the induction of demyelinating MS-like lesions.

#### 4.1.3. Mechanisms of HCoV-Induced Neurodegeneration: Possible Associated Neuropathologies

Immune cell infiltration and cytokine production were observed in the mouse CNS after infection by HCoV-OC43. This immune response was significantly increased after infection by viral variants, which harbor mutations in the viral glycoprotein (S) [267]. These variants also induced glutamate excitotoxicity [268,289], thus increasing damage to neurons [310] and/or disturbing glutamate homeostasis [311] and thereby contributing to neuronal degeneration and hind-limb paralysis and possible demyelination [266,267,268,269]. The degeneration of neurons may eventually lead to death of these essential cells by directly generating a cytotoxic insult related to viral replication and/or to the induction of different regulated cell death (RCD) pathways [312,313,314]. Our results indicate that the underlying mechanisms appear to involve different cellular factors and pathways of RCD, described and reviewed elsewhere [38,315].

Virus–cell interactions are always important in the regulation of cell response to infection. For HCoV-OC43, we clearly showed that the viral S and E proteins are important factors of neurovirulence, neuropropagation and neurodegeneration of infected cells [267,268,269,296,312]. We have also demonstrated that the HE protein is important for the production of infectious HCoV-OC43 and for efficient spreading between neuronal cells, suggesting an attenuation of the eventual spread into the CNS of viruses made deficient in fully active HE protein, potentially associated with a reduced neurovirulence [269,295]. Coronavirus accessory proteins have been extensively studied and are now considered as important viral factors of virulence implicated in pathogenesis while counteracting innate immunity [316,317,318,319]. Two of these accessory proteins (ns2 and ns5) produced during infection by HCoV-OC43 play a significant role in virulence and pathogenesis in the mouse CNS [38].

Like for several other respiratory viruses, accumulating evidence now indicate that HCoV are neuroinvasive in humans and we hypothesize that these recognized respiratory pathogens are potentially neurovirulent as well, as they could participate in short- and long-term neurological disorders either as a result of inadequate host immune responses and/or viral propagation in the CNS, which directly induces damage to resident cells. With that in mind, one can envisage that, under the right circumstances, HCoV may successfully reach and colonize the CNS, an issue largely deserted and possibly underestimated by the scientific community that has impacted or will impact the life of several unknowing individuals. In acute encephalitis, viral replication occurs in the brain tissue itself, possibly causing destructive lesions of the nervous tissue with different outcomes depending on the infected regions [320]. As previously mentioned, HCoV may persist in the human CNS as it does in mice [245,287] and potentially be associated with different types of long-term sequelae and chronic human neurological diseases. 

## 5. Conclusions

In their famous review on CNS viral infection, published a few years ago, Koyuncu et al. [35] insisted that, under the right conditions, all viruses can have access to the CNS. What “under the right conditions” means certainly represents a subject of debate among virologist and physicians. Nevertheless, as stated in the introduction of this review, viral factors (mutations in specific virulence genes), host factors (immunodepression, age) or a mixture of both (underlining the importance of virus–host interactions), are all good candidates to refer to if one intends to find the beginning of an explanation. A fast and accurate diagnosis would certainly improve prognosis for patients with a suspected CNS infection. Identification of a specific virus provides relevant information on how to treat a patient; therefore, the development of modern technologies, such as high throughput sequencing (Next Generation Sequencing) are warranted as it represents a potentially unbiased marvelous tool for rapid and robust diagnosis of unexplained encephalitis or other types of encephalopathies or neuronal manifestations, especially in the context where more traditional techniques have failed to identify the etiological agent [21,108,111,244,321,322]. Therefore, although our attention is mainly on a few different viruses such as HSV, arboviruses and enteroviruses, it may now be the time to look at CNS viral infection from another perspective. These viruses truly represent an important proportion of CNS viral infection associated with encephalitis, meningitis, myelitis and long-term neurological disorders. Nevertheless, accumulating evidence in the scientific literature strongly suggest that many other viral candidates could be underestimated in that matter. 

Several human respiratory viruses are neuroinvasive and neurotropic, with potential neuropathological consequences in vulnerable populations. Understanding the underpinning mechanisms of neuroinvasion and interaction of respiratory viruses (including HCoV) with the nervous system is essential to evaluate potentially pathological short- and long-term consequences. However, viral infections related to diseases that are rare manifestations of an infection (like long term chronic neurological diseases), represent situations where Koch’s postulates [323] need to be modified. A series of new criteria, adapted from Sir Austin Bradford Hill, for causation [324,325] was elaborated by Giovannoni and collaborators concerning the plausible viral hypothesis in MS [326]. These criteria certainly represent a pertinent tool to evaluate the involvement of human respiratory viruses as a factor that could influence long-term human neurological diseases. To continue the gathering of epidemiological data is justified to evaluate the clear cause and effect link between neuroinvasive respiratory viruses and short- and long-term human neurological diseases. Understanding mechanisms of virus neuroinvasion and interactions with the central nervous system is essential for different reasons. First, to help better understand potentially pathologically relevant consequences of infection, and second in the design of novel diagnostic and intervention strategies that will help uncover potential “druggable” molecular virus–host interfaces highly relevant to symptoms of various neurological diseases with a viral involvement.

## Figures and Tables

**Figure 1 viruses-12-00014-f001:**
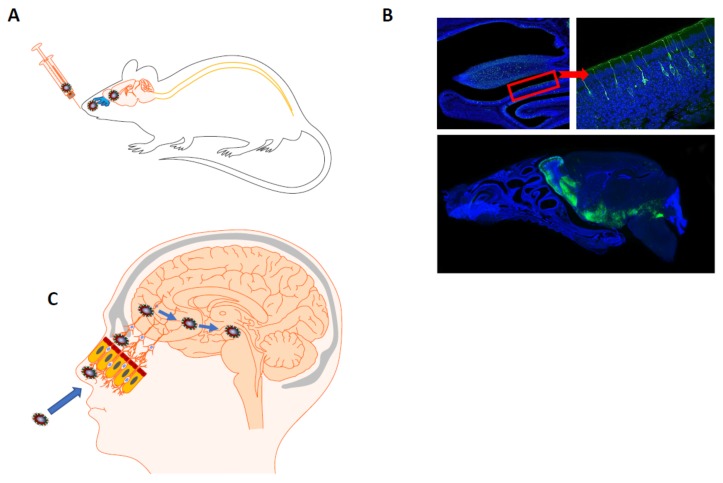
Illustration of the principal route of infection used by HCoV-OC43 for neuroinvasion in the central nervous system (CNS). (**A**) Schematic representation of intranasal injection of HCoV-OC43 in susceptible mice. (**B**) Histological examination of decalcified whole head allows to visualize virus spread in the CNS at 3 dpi. **Left top panel** represents the nasal cavity and **right top panel** represents a higher magnification of infected olfactory receptor neurons (ORN) in the neuroepithelium. **Bottom panel** represents viral dissemination in several regions of the brain from the olfactory bulb to the brainstem. The inset on the right represents a zoomed image of the area delimited by the red frame in left panel. The red arrow indicates the enlarged region in the red frame. In all regions of the brain, neurons are the main target of infection. Detection of viral S glycoprotein (green) and cell nucleus (DAPI; blue). Magnification is 20× and 63× for **upper panels** and 4× for the **bottom panel**. (**C**) Corresponding schematic representation of intranasal infection in humans. HCoV may infect the ORN, pass through the neuroepithelium and gain access to the olfactory bulb (OB) and eventually to other regions of the brain. The blue arrows indicate the direction of viral spreading. Schematic representations were assembled using the Motifolio Neuroscience Toolkit 2007.

**Figure 2 viruses-12-00014-f002:**
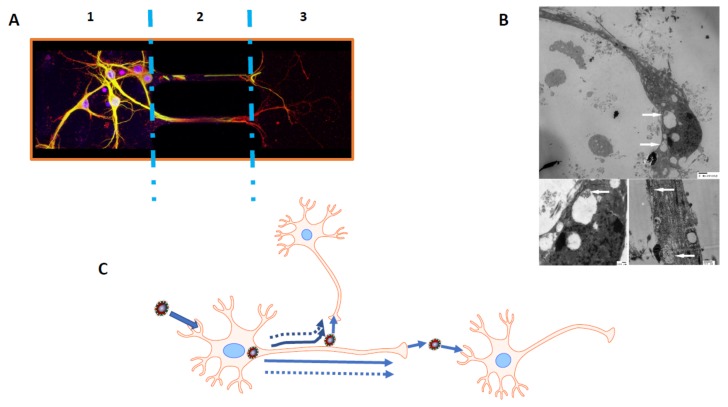
Model of axonal transport and neuron-to-neuron or neuron-to-non-neuronal cells propagation. (**A**) Murine primary mixed neuronal cultures (PMNC) grown in Xonachip microfluidic compartmentalized chambers. These devices allow fluidic isolation of axons by establishing a volume difference between cell bodies (1) and axonal end (3) compartments and the high fluidic resistance of the microchannels (2) (where the axons grow) produces a sustained flow that counteracts diffusion. The blue dotted lines are there to separate the three different parts of the microfluidic device. This situation blocks the migration of free virus through the microchannels and makes it possible to use this system in viral infection studies in neuronal cultures. PMNC were infected in the cell body portion (1) and viral S glycoprotein (yellow/green) and neuronal marker (MAP2 protein; red) detection was performed on fixed cells at 24 hpi. Data are indicative of viral antigens and/or viral particles going from cell body (1) through the axon in the microchannel (2) and then towards the axonal end portion (3). (**B**) Electron microscopy images of infected PMNC grown on Aclar-33C embedding film (Electron Microscopy Sciences) and infected at an MOI of 0.03 for 48 h at 37 °C. Sliced embedded samples (EPON RESIN 828; Polysciences Inc., Warrington, PA, USA) observed with a Hitachi H 7100 electron microscope show viral particles (white arrows) in both cell bodies and axons at 48 hpi. **Upper panel** is a complete neuron (magnification 5000×), **left lower panel** is a representative neuronal cell body (magnification 10,000×) and **lower right panel** is a representative axonal portion (magnification 20,000×). Pictures were taken with an AMTXR-111 camera (Advanced Microscopy Techniques, Woburn, MA, USA). (**C**) Model of HCoV-OC43 propagation (neuron-to-neuron or neuron-to-non-neuronal cells) based on our data [269] and adapted from Tomishima and Enquist [291]. In this model, solid arrows represent fully assembled virus transport and dashed arrows represent subvirion assemblies [291]. Schematic representations were assembled with the Motifolio Neuroscience Toolkit, 2007.

**Figure 3 viruses-12-00014-f003:**
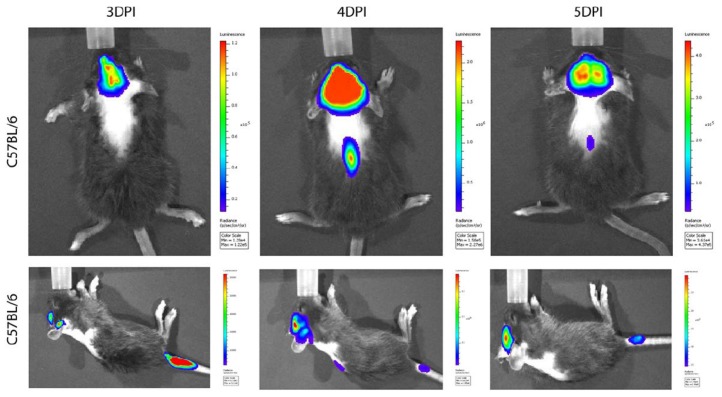
Non-invasive imaging of viral neuroinvasion and dissemination within the CNS in living infected mice and associated clinical scores. A recombinant HCoV-OC43 harboring a luciferase reporter gene [292] was injected intra-nasally (I.N.) into mice. Virus spread was assessed by bioluminescence imaging (BLI) with the Xenogen VIVO Vision IVIS 100 imaging system (Perkin-Elmer) in infected anaesthetized mice placed in a light proof specimen chamber after intraperitoneal injection of d-luciferin. Images were taken with a CCD camera mounted in a light-tight imaging chamber, using the acquisition software Living Image version 4.3.1 (Caliper-LifeSciences). Evaluation of associated clinical scores: (levels 0 to 4: 0 is asymptomatic; 1 is mice with early hunched back; 2 is mice presenting slight social isolation, weight loss and abnormal gait; 3 is mice presenting total social isolation, ruffled fur, hunched back, weight loss and almost no movement; and 4 is mice moribund or dead (presented elsewhere; [266]), indicate that only mice with a positive signal at both the level of the brain and spinal cord were evaluated to be at level 2 to 3.

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
