# Peer review of "Human Coronaviruses and Other Respiratory Viruses: Underestimated Opportunistic Pathogens of the Central Nervous System?"

_viruses, 2019, doi:10.3390/v12010014_

Round 1

Reviewer 1 Report

This review highlights what is known and the information gaps in our understanding of the neurological complications associated with respiratory virus infections. It is generally well written and suitable for the journal. 

Major criticism.

My major criticism is that it is clearly missing a section, either immediately after or within the introduction, that explains the neuro-innervation of the airway and the other routes by which respiratory viruses cross over to the CNS. They mention aspects of this sporadically throughout the manuscript but an article such as this one begs for a section on how the airways and CNS are connected.

Minor criticism.

The grammar and spelling need some editing. I have provided a marked up version of the errors that I have found but the authors will also have to give the manuscript a thorough edit before resubmission of their revisions. The markups can be viewed on Adobe reader.

Author Response

Responses to reviewers

Reference : Viruses-654080

Title : Human Coronaviruses and Other Respiratory Viruses: Underestimated Opportunistic Pathogens of the Central Nervous System?

Authors :   Marc Desforges, Alain Le Coupanec, Philippe Dubeau, Andréanne Bourgouin, Louise Lajoie, Mathieu Dubé, and Pierre J. Talbot

REVIEWER #1

General comments:

Comments and Suggestions for Authors

This review highlights what is known and the information gaps in our understanding of the neurological complications associated with respiratory virus infections. It is generally well written and suitable for the journal. 

Major criticism.

My major criticism is that it is clearly missing a section, either immediately after or within the introduction, that explains the neuro-innervation of the airway and the other routes by which respiratory viruses cross over to the CNS. They mention aspects of this sporadically throughout the manuscript but an article such as this one begs for a section on how the airways and CNS are connected.

Minor criticism.

The grammar and spelling need some editing. I have provided a marked up version of the errors that I have found but the authors will also have to give the manuscript a thorough edit before resubmission of their revisions. The markups can be viewed on Adobe reader.

RESPONSE TO REVIEWER 1

We first want to thank the reviewer for our "5 stars" evaluation on most criteria in the review report.

Response to Major criticism: We agree with the reviewer about this comment and we have now introduced a new paragraph in section 2 of the manuscript. This paragraph contains portions of the text that were already present at different places in the “old version” of the manuscript and we completed it with more information to describe the different points that were suggested by the reviewer.

Response to Minor criticism: Sorry for the quality of the language in the previous version of the manuscript. We greatly thank the reviewer for his or her help in the edition of the text. We made a thorough edition and we now think that the manuscript is greatly improved. 

The new version of the manuscript is in attachment.

Marc Desforges PhD.

Reviewer 2 Report

In the abstract, the first line should read "Several respiratory viruses infect the human upper respiratory . . . " The sentence in the abstract starting with "Given that the etiological agent . . ." (lines 31-35) is confusing and needs to be reworded.  I believe the authors are simply trying to say that the role of opportunistic pathogens needs to be considered for viral-related neurological disorders. In the introduction (line 45) the authors mention "occurring alterations" but it is not clear what is intended here.  It is difficult to determine if the authors are referencing changes to the CNS or to the viruses themselves. In the introduction (line 48) the authors mention that the incidences of viral infection of the CNS in clinical practice is difficult "to precise".  I believe they intended to say that their incidence in clinical practice is "difficult to determine precisely".   The sentence in the introduction explaining the prevalence of viral encephalitis (lines 48-51) is not clear and needs to be reworded.  It is unclear what point the authors are trying to make other than it is difficult to definitively detect viruses in cases of viral encephalitis. In the introduction (line 52) change the word "requiring" to "which require" to make the sentence easier to read.   In the introduction (lines 55-57) consider rewording that sentence to increase clarity.  Perhaps something along the lines of "Due to the cost associated with patient care/treatment, CNS viral infection cause increased morbidity and more severe disabilities in low-income/resource-poor countries." In the introduction (lines 62-65) it unclear what the authors are referencing for the other components of respiratory disease.  Is the idea that viruses make up 95% of respiratory disease in children and 30-40% in the elderly while fungal and bacterial infections account for the rest?   In the introduction (line 72) replace "is crossing" with the word "crosses".   In section 2 (line 77) the sentence should read "Most of these infections are self limited. . ." In section 2 (line 87), the sentence should read "Although the CNS seems difficult for viruses to infect, viruses able to travel within the CNS will be able . . . " In section 2 (line 101), change the start of the sentence to be "Symptoms may include:" rather than "When present, symptoms may be:" In section 2 (line 109), the authors need to clarify that they are discussion CNS resident cells. In section 3 (line 162), cases is plural so "has" should be "have" In section 3 (line 198), reference 118 discusses CNS complications from influenza B virus, not influenza A virus. In section 3 (lines 215-220), the authors should separate EV-A71 and EV-D68 into their own sentences.  While EV-A71 can be found in the respiratory system, it is not associated with respiratory disease and it is difficult to tell which virus is tied to which diseases when they are discussed in the same sentence. In the heading for section 4.1 (line 275) consider rewording for clarity. "Pathogens causing also extra-respiratory illnesses" is difficult to understand. In section 4.1.1 (lines 327-328) remove the word "really" as it is not quantifiable.  This sentence could be simplified to "Although the olfactory bulb is effective at controlling neuroinvasion, several viruses . . ."  In section 4.1.1 (lines 341-345) the authors state that HCoV may be a factor or co-factor for long-term neurological sequelae.  This sentence seems out of place since the research supporting this conclusion are addressed in the sections after this.  Consider moving this sentence later in the manuscript.

Author Response

REVIEWER #2

In the abstract, the first line should read "Several respiratory viruses infect the human upper respiratory . . . " The sentence in the abstract starting with "Given that the etiological agent . . ." (lines 31-35) is confusing and needs to be reworded.  I believe the authors are simply trying to say that the role of opportunistic pathogens needs to be considered for viral-related neurological disorders. In the introduction (line 45) the authors mention "occurring alterations" but it is not clear what is intended here.  It is difficult to determine if the authors are referencing changes to the CNS or to the viruses themselves. In the introduction (line 48) the authors mention that the incidences of viral infection of the CNS in clinical practice is difficult "to precise".  I believe they intended to say that their incidence in clinical practice is "difficult to determine precisely".   The sentence in the introduction explaining the prevalence of viral encephalitis (lines 48-51) is not clear and needs to be reworded.  It is unclear what point the authors are trying to make other than it is difficult to definitively detect viruses in cases of viral encephalitis. In the introduction (line 52) change the word "requiring" to "which require" to make the sentence easier to read.   In the introduction (lines 55-57) consider rewording that sentence to increase clarity.  Perhaps something along the lines of "Due to the cost associated with patient care/treatment, CNS viral infection cause increased morbidity and more severe disabilities in low-income/resource-poor countries." In the introduction (lines 62-65) it unclear what the authors are referencing for the other components of respiratory disease.  Is the idea that viruses make up 95% of respiratory disease in children and 30-40% in the elderly while fungal and bacterial infections account for the rest?   In the introduction (line 72) replace "is crossing" with the word "crosses".   In section 2 (line 77) the sentence should read "Most of these infections are self limited. . ." In section 2 (line 87), the sentence should read "Although the CNS seems difficult for viruses to infect, viruses able to travel within the CNS will be able . . . " In section 2 (line 101), change the start of the sentence to be "Symptoms may include:" rather than "When present, symptoms may be:" In section 2 (line 109), the authors need to clarify that they are discussion CNS resident cells. In section 3 (line 162), cases is plural so "has" should be "have" In section 3 (line 198), reference 118 discusses CNS complications from influenza B virus, not influenza A virus. In section 3 (lines 215-220), the authors should separate EV-A71 and EV-D68 into their own sentences.  While EV-A71 can be found in the respiratory system, it is not associated with respiratory disease and it is difficult to tell which virus is tied to which diseases when they are discussed in the same sentence. In the heading for section 4.1 (line 275) consider rewording for clarity. "Pathogens causing also extra-respiratory illnesses" is difficult to understand. In section 4.1.1 (lines 327-328) remove the word "really" as it is not quantifiable.  This sentence could be simplified to "Although the olfactory bulb is effective at controlling neuroinvasion, several viruses . . ."  In section 4.1.1 (lines 341-345) the authors state that HCoV may be a factor or co-factor for long-term neurological sequelae.  This sentence seems out of place since the research supporting this conclusion are addressed in the sections after this.  Consider moving this sentence later in the manuscript. 

RESPONSE TO REVIEWER 2

Sorry for the quality of the language in the previous version of the manuscript. We greatly thank the reviewer for his or her help in the edition of the text. We made a thorough edition and we now think that the manuscript is greatly improved as we made the majority of the changes according to the reviewer’s suggestions.

However, even though we understand the reviewer’s point of view about the sentence at the end of section 4.1.1, we feel that this sentence makes a good transition to the next section and we decided to keep this portion intact.

Our new manuscript is in attachment.

Regards,

Marc Desforges PhD.

This manuscript is a resubmission of an earlier submission. The following is a list of the peer review reports and author responses from that submission.